# From Macro- to Microscale: A combined modelling approach for near-surface wind flow on Mars at sub-dune length-scales

**Richard Love**[1]*, **Derek W. T. Jackson**[1,2], **Timothy Michaels**[3], **Thomas A. G. Smyth**[4], **Jean-Philippe Avouac**[1,5], **Andrew Cooper**[1,2]

**1** School of Geography & Environmental Sciences, Ulster University, Northern Ireland, United Kingdom, **2** Geological Sciences, University of KwaZulu-Natal, Durban, South Africa, **3** Carl Sagan Center (at the SETI Institute), Mountain View, California, United States of America, **4** Department of Biological and Geographical Sciences, School of Applied Sciences, University of Huddersfield, England, United Kingdom, **5** Division of Geological and Planetary Sciences, CalTech, Pasadena, California, United States of America

* love-r3@ulster.ac.uk

**Data Availability Statement:** Data has been uploaded to the following data repository: https://osf.io/wkp3c.

## Abstract

The processes that initiate and sustain sediment transport which contribute to the modification of aeolian deposits in Mars' low-density atmosphere are still not fully understood despite recent atmospheric modelling. However, detailed microscale wind flow modelling, using Computational Fluid Dynamics at a resolution of <2 m, provides insights into the near-surface processes that cannot be modeled using larger-scale atmospheric modeling. Such Computational Fluid Dynamics simulations cannot by themselves account for regional-scale atmospheric circulations or flow modifications induced by regional km-scale topography, although realistic fine-scale mesoscale atmospheric modeling can. Using the output parameters from mesoscale simulations to inform the input conditions for the Computational Fluid Dynamics microscale simulations provides a practical approach to simulate near-surface wind flow and its relationship to very small-scale topographic features on Mars, particularly in areas which lack *in situ* rover data. This paper sets out a series of integrated techniques to enable a multi-scale modelling approach for surface airflow to derive surface airflow dynamics at a (dune) landform scale using High Resolution Imaging Science Experiment derived topographic data. The work therefore provides a more informed and realistic Computational Fluid Dynamics microscale modelling method, which will provide more detailed insight into the surface wind forcing of aeolian transport patterns on martian surfaces such as dunes.

## Introduction

The dominant mode of contemporary surface geomorphological change on Mars is through the transport and deposition of sand by aeolian processes [1–3]. The near-surface atmospheric density of Mars is around 0.02 kg/m³, ~100 times less dense than that on Earth. This low-density atmosphere requires much higher wind speeds, up to 7 times higher than on Earth, to initiate sediment transport [2,4]. However, once initiated, the wind speeds required to maintain

**Funding:** The author(s) received no specific funding for this work.

**Competing interests:** The authors have declared that no competing interests exist.

transport are much lower [5–7]. Despite the low-density Martian atmosphere, the lower gravity ($g = 3.71$ ms$^{-2}$) increases the length of time a sediment particle can be accelerated by near-surface winds and lifted into the air, where they are lifted higher and for longer trajectories than on Earth [8]. When the particle falls, it has sufficient energy upon impact with the surface to initiate the lifting of other grains into the air, which continues the process. This causes a chain effect called a 'saltation cluster' [5]. This process allows for sediment transport to occur on Mars even when the wind friction velocity falls below the fluid threshold wind speed but remains above the impact threshold wind friction speed. There is a wealth of evidence that sediment transport and dune migration does occur on Mars [9]. Dust accumulation on the Spirit and Opportunity Rovers [8,10], observations of dust movement from the Mars Orbiter Camera onboard the Mars Global Surveyor [11], movement of dunes within the Bagnold dune field [12], sediment movement within Proctor crater [13–15], dune activity in Herschel Crater [16] and sand fluxes in Nili Patera [17–19] are among the examples of this observed sediment transport. Evidence of sediment transport is, however, inconsistent with the fact that winds capable of sediment transport have only rarely been observed or inferred/simulated [5,7,8,20–22]. The discrepancy between the observed sediment transport and the apparent absence of frequent wind speeds capable of sediment transport might be resolved by further research into microscale wind processes using CFD modelling.

In lieu of a wide range of high quality *in situ* data from Mars [23], research examining atmospheric surface-interactions on Mars and their resultant geomorphological landform changes, defined here as microscale atmospheric modelling, normally relies on numerical simulations of the atmosphere [4,15]. The meteorological data collected from landers/rovers (Viking 1 and 2, Pathfinder, Phoenix, Curiosity, InSight and Perseverance), while important to our understanding of atmospheric conditions, are restricted to a very small portion of the surface of Mars and only the Viking landers were equipped with appropriate wind speed and direction sensors [24]. The calibration of Pathfinder wind measurements proved more difficult than anticipated and was never completed. The Phoenix lander used a simple wind sensor (the Telltale) that had relatively low measurement cadence and accuracy. Upon landing on Mars, Curiosity's wind sensors were found to be seriously damaged, curtailing rigorous collection of wind data. InSight has wind sensors, however the sensor's relatively low height and proximity to other instrumentation such as solar panels limits the use of collected data from a microscale meteorological perspective [25,26]. The wind measurements from Perseverance are faced with similar self-contamination issues. Regardless, the limited number of lander/rover missions to Mars, and most importantly, the limited sampling of different terrains and regions by those spacecraft means that the *in situ* data that has been collected is of limited direct use elsewhere on the planet. Computational Fluid Dynamics (CFD) modelling can be used to examine near-surface microscale atmospheric processes in areas which are of geomorphological interest, but lack *in situ* data. Simulations at multiple atmospheric scales, utilising the output of the preceding scale to inform the next level will provide a more realistic approach to performing CFD microscale modelling. The process of acquiring the input data for CFD simulations was carried out by initially simulating macroscale conditions using a Global Climate Model (GCM), with the output from the GCM simulation being used to drive the regional fine-scale mesoscale simulations, and finally the output from the mesoscale simulations was then used to inform the boundary conditions for the CFD simulation.

GCMs (also known as Global Circulation Models) provide insights into large-scale processes in the Martian atmosphere [20,27–36]. The resolution of these GCMs, however, is typically much too coarse ($\Delta \sim 100$–$500$ km resolution) to adequately understand the processes and interactions at less than a dune field length scale [37]. They are also not capable of providing realistic representations of important smaller-scale circulations on Mars [38]., and thus

provide limited insight into transport processes in the lower atmosphere [39,40]. The surface of Mars is covered with numerous wind-modulating topographic features that are too small to be accounted for in GCMs [37], but are too large to be properly treated in CFD simulations. For example, the multiple dune fields of Gale crater (about 150 km in diameter) are acted upon by multiple wind directions (greatly modulated by the crater's rim and central mound) that could not be properly resolved by the one or even several GCM grid points that would cover the crater. Therefore, the output from a GCM simulation should not be directly used as the inlet conditions for most microscale CFD modelling over bedforms. In order to model the wind flow processes which may contribute to the modification of aeolian landforms on Mars, higher-resolution simulations of the atmosphere are required.

Regional mesoscale models are supplied with an initial state and time-dependent three-dimensional boundary conditions from the output of larger-scale GCM simulations [23]. In this study, we develop a set of protocols for the process of informing the boundary conditions of the CFD simulations performed on a microscale level, using the output from a mesoscale model which has a resolution ranging from of tens of meters to hundreds of kilometers depending on the application (itself informed by a GCM). Mesoscale simulations help account for the role of complex regional topography on atmospheric winds [39,41], which provides more insight into understanding the role of mesoscale processes on microscale near-surface wind behavior [16,42–44]. Successes in reproducing *in situ* meteorological data from landed spacecraft using GCM and mesoscale climate modelling [31,37,42,45–47] have provided confidence in using mesoscale simulation output to inform CFD microscale simulations. The validation of large-scale modelling simulations using *in situ* measurements underscores the importance that *in situ* data can have on our understanding of processes on Mars. This study aims to develop a procedure to examine microscale conditions at sites on Mars which do not have the benefit of such measurements. Despite great improvements in the capability of mesoscale climate modelling, the resolution of the model output at this scale is still much coarser than the scale of aeolian bedforms it is often used to interpret, and therefore still falls short of the high resolution required for analysis of near-surface microscale wind flows and associated geomorphological changes for which high-resolution topographic data exist—hence the use of CFD modeling to partially bridge this gap.

There has been increased interest in examining microscale wind flow and associated geomorphological processes on Mars. The effect of local dune topography on the steering and modification of ripples was examined using a ~5 m CFD airflow model resolution within Proctor crater [15]. CFD simulations were also used to investigate dune slip face dynamics on Mars, with a 1 m surface airflow model resolution using a High Resolution Imaging Science Experiment (HiRISE) Digital Elevation Map (DEM) [4]. Seasonal wind patterns over the Bagnold dunes were examined [4], which provided new insights into sediment transportation which could not be resolved by mesoscale modelling alone. However, there are only a limited number of studies which have used CFD simulations to examine near-surface microscale flow dynamics on Mars at sub-dune length scales. Although a large number of combined modelling approaches have been developed and applied to terrestrial-based microscale modelling [48–53], these models are distinctly different to each other and a single method of combining multiple scales has yet to be adopted for Earth. Furthermore, none of the approaches which have been successfully implemented on Earth can be directly applied to the vastly different physical properties of the Martian atmosphere ($CO_2$ ice and water ice in the atmosphere, lower gravity, lower pressure, lower albedo, lower temperatures) which require Mars-specific models. An example of a specific model made for Mars-based atmospheric processes is the mesoscale model used in this study, the Mars Regional Atmospheric Modelling System (MRAMS). MRAMS was originally based on the Earth-based model RAMS (Regional Atmospheric

Modelling System), but over time elements of the model were modified, including the model set-up, boundary conditions, model physics, nested grid setup, and details of the model core [54].

Therefore, a new approach is required to examine the microscale surface-atmospheric interactions over meter- and decameter-scale topography on Mars. The progress made on similar Earth-based modelling is in stark contrast to the lack of developments made to examine the Martian atmosphere using a combined modelling approach. While mesoscale simulations of the Martian atmosphere rely on larger-scale modelling (GCM) output to inform their initial state and time-dependent boundary conditions, microscale studies on Mars have relied on multiple sources of data: using input data acquired through remote sensing [15,55], the limited *in situ* data collected by lander missions [4,56], simulating microscale conditions using a mesoscale model at very high resolution, or performing partially-idealized Large Eddy Simulations (LES; e.g., [23,57,58]. This study details a new multi-scale modelling approach which uses the macroscale output from a fine-scale (typically 1 to 10 km grid spacing, depending on the topographic setting) mesoscale model to provide a more realistic method for informing the boundary conditions of CFD microscale simulations of near-surface wind dynamics on Mars, particularly for areas which lack *in situ* atmospheric data. We aim to provide a new set of protocols to perform microscale CFD modelling at a sub dune length scale. This requires using the output of a GCM to inform mesoscale modelling over the area of interest, the output of which will be used to inform the initial state and boundary conditions of the CFD model. This process is outlined in Fig 1, which provides a general overview of the methodological process used for this study.

Alongside the modelling workflow shown in Fig 1, the process of site selection is shown. While the region of interest can be selected from low resolution imagery, the selected site must also have temporal HiRISE imagery. The high resolution imagery from the HiRISE camera is vital for the creation of a Digital Terrain Model (DTM) which is used to create the stereolithography (STL) file. The STL file provides realistic dune topography within the CFD domain, which is crucial to properly examine wind flow behaviour over the site of interest.

The proposed method was tested by simulating near-surface wind flow over a barchan dune field within the Nili Patera caldera, which we find was able to provide insights into the microscale aeolian forcing conditions contributing to the morphology of the dunes under investigation.

## Global circulation model

This study used output from a contemporary simulation of the NASA/Ames GCM. The model was described in detail in [30], and only a summary is presented here. This GCM is a hydrostatic model, it prescribes the atmospheric dust loading and computes the dust and water cycle (including water- ice-cloud microphysics with nucleation, growth and settling of water ice) [30]. It also incorporates a radiatively-active cloud system, which strengthens the role of large-scale atmospheric circulation. Mars Orbiter Laser Altimetry (MOLA) data is used to calculate the surface topography, based off the 1/16[th] degree MOLA topography in [59], which is then smoothed to the 5˚x6˚ grid scale used in this version of the model, utilising a simple cylindrical map projection for the horizontal grid. The NASA/Ames GCM used in this study uses a sigma-pressure vertical grid that generally follows the surface topography near the surface. A uniform surface roughness length of 0.01 m is used. The planetary boundary layer is accounted for within this model using diffusion for turbulent mixing dependent on the Richardson number, described further in [45,60]. The output fields from the NASA/Ames GCM simulation are used to prepare the initial state and time-dependent boundary conditions needed by the

## Workflow

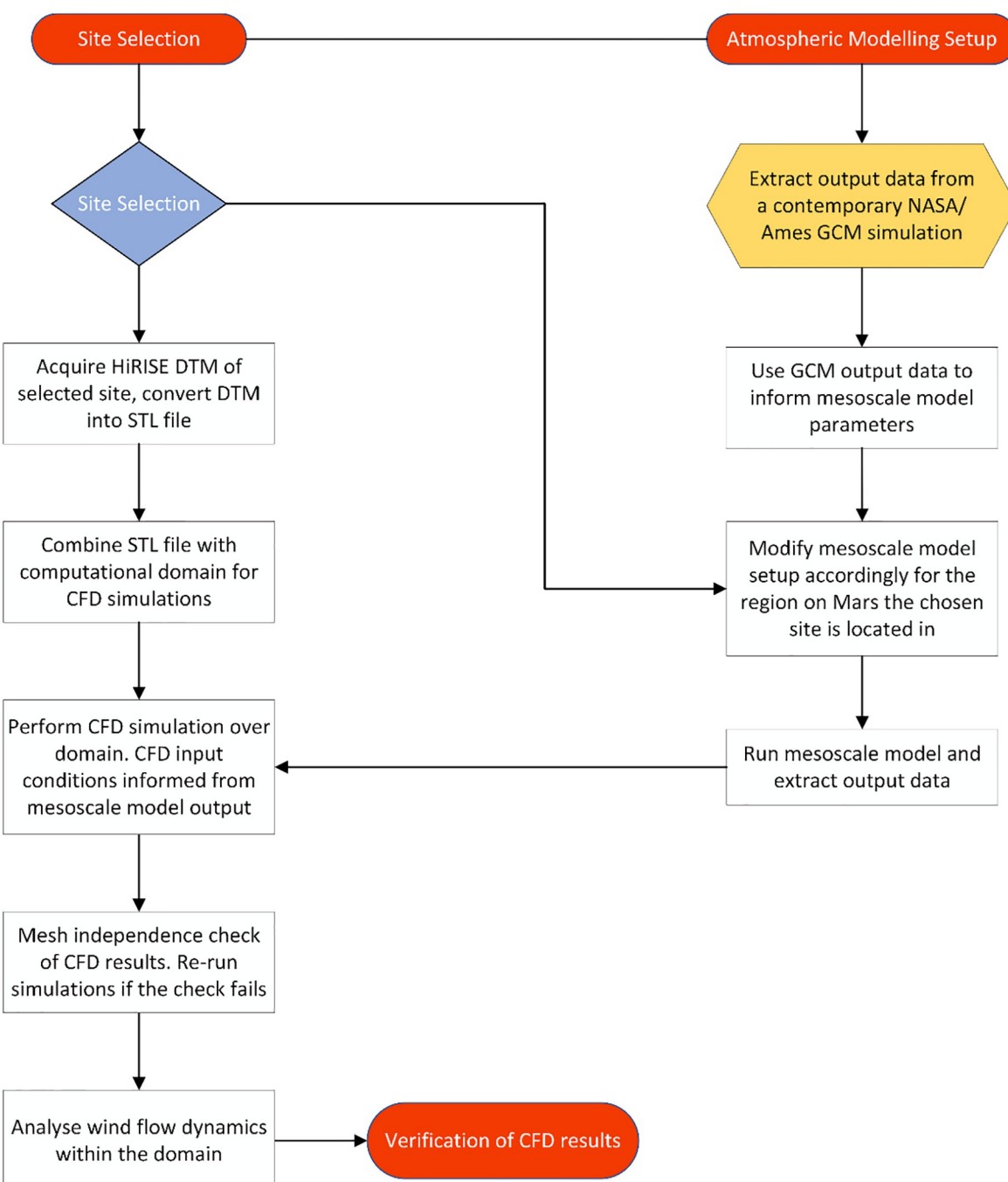

**Fig 1. Workflow showing the multi-scale modelling approach used to generate localised Computational Fluid Dynamics results of surface airflow.**

mesoscale model. Details of this type of pre-processing, in order to successfully implement GCM data as the initial state for MRAMS simulations can be found in published work, [37,42,54]. The output data from the GCM used as input into the mesoscale model for this study include but are not limited to: wind velocities, surface, sub-surface and atmospheric temperatures, atmospheric pressure, $H_2O$ ice and $CO_2$ surface ice volumes, dust concentration, water vapor concentration and cloud mass. The Mars Year (MY) used in the GCM for this study was MY 24 which is a combination of the conditions occurring during Ls 0˚ to ~100˚ (MY 25–26) and observations from Ls ~100˚ to 360˚ of MY 24. MY 24 was a year which did not have an observed global dust storm and it is therefore seen to be an appropriate representation of a 'typical' MY [30].

## Mesoscale model

MRAMS is a non-hydrostatic model, developed from RAMS, a model used for Earth [37]. MRAMS simulations were run in this study using the output data from the NASA/AMES MGCM as the initial state and time-dependent boundary conditions. MRAMS includes the effect of atmospheric dust on radiative processes [54]. The radiative transfer scheme in this simulation is a two-stream radiative transfer code based on the Community Aerosol and Radiation Model for Atmospheres implementation [61]. The MRAMS set-up simulated in this study has a finest horizontal resolution of 2 km and the atmospheric layer nearest the surface is 10 m thick. This spatial resolution is fine enough in order to account for the effect of most complex topography near Nili Patera on flows, including anabatic and katabatic winds. These winds are caused by diurnal variations in local surface temperatures and the surrounding atmospheric temperatures on sloping terrain [24]. Anabatic winds are winds that flow upslope during the day as a result of solar heating [24] while katabatic winds are nighttime winds where cool, dense air, flows downslope accelerated by gravity [62]. The significant radiative cooling/heating of both the atmosphere and surface of Mars enhance the production of these winds [63]. These diurnal winds have significantly different properties and ultimately influence the ability for sediment transport to be induced and therefore must be accounted for by the mesoscale modelling.

A MRAMS simulation was run for this study, ~4.5 sols in duration at Ls ~270˚. A sol is a solar day on Mars, with a mean period of ~24 hours 39 minutes. The model was set up with an output interval of 20 Mars-minutes (each Mars-minute is 1/1440 of a sol). The model 'spins up' [64] as it performs initial calculations of the atmospheric processes, therefore the first sol of data from the mesoscale output is discarded, as the data calculated during this period is unreliable. Although MRAMS was run for 4.5 sols, only 3.5 sols of output from the mesoscale simulation are used to inform the input to the CFD simulation. The surface roughness value for the MRAMS run was set to 0.01 m, based roughly on the map of [65]. The model topography is self-calculated using the gridded 1/128˚ MOLA dataset, with the topographic value for each grid point calculated as an average of the MOLA values within the grid cell. MRAMS was run using an interactive nested grid system, which is used to increasingly focus the resolution of the model from a hemispheric scale down to the region of interest using five nested domains in total. The MRAMS vertical levels follow the terrain topography near the surface and become increasingly horizontal as height increases above the surface. The output from the MRAMS simulation that were used to inform the boundary conditions of the CFD simulations included the u, v and w wind components within the first kilometer of the atmosphere, friction velocity, and atmospheric density.

Grid 1, the coarsest grid of the model, had a horizontal grid spacing of ~240 km, while the highest resolution of the interactive grids (grid 5) had a ~2 km horizontal grid spacing (Fig 2).

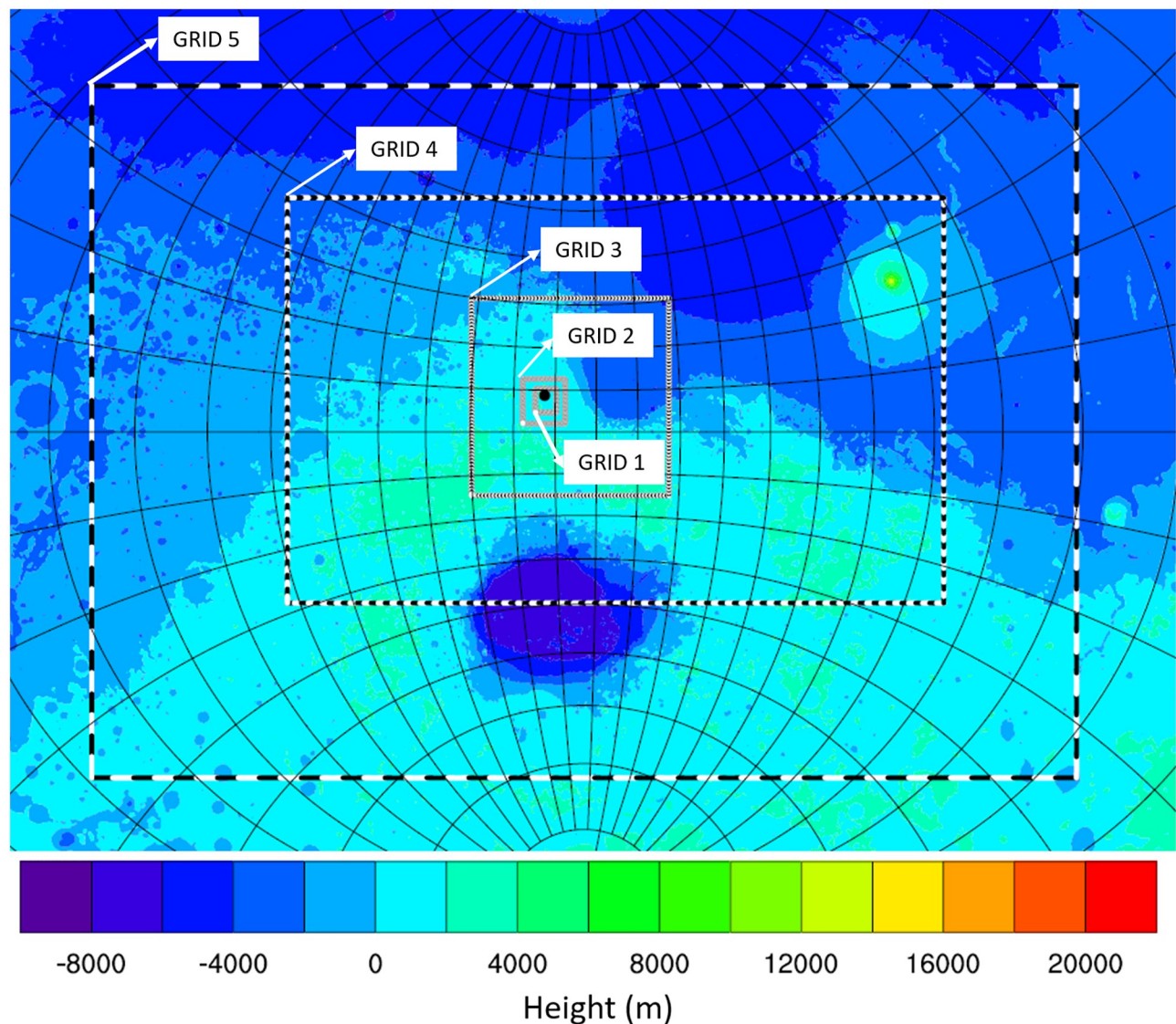

**Fig 2. Showing the MRAMS grid setup, the black mark in Grid 1 shows the location of Nili Patera.**

This is close to the 2.96 km and 1.5 km grid spacings utilised by [18,64] who employed 1.5 km grid spacing for the innermost nest of their Mars Weather and Researching Forecasting Model (MarsWRF) mesoscale simulation. MRAMS vertical grid spacing used a constant geometrical stretching factor defined as the ratio of the grid spacing between adjacent vertical levels [37]. The vertical grid spacing increases with height above the surface. The model had a spacing of 10 m at the surface (the first vertical layer) increasing to a maximum layer thickness of 2000 m. The vertical model domain was ~50 km thick, discretized into 70 vertical layers.

## Worked example

The site for this study is the Nili Patera dune field (8.8°N, 67.3E°), a large barchanoid dune field (~200 km$^2$). The dune field is located within Nili Patera caldera, a 55 km wide, ~2 km deep caldera within Syrtis Major Planum [18], (Fig 3a and 3b). The area upwind of the dunes of interest (north-east portion of the dune field) is relatively flat (Fig 3c), over 10 km from the

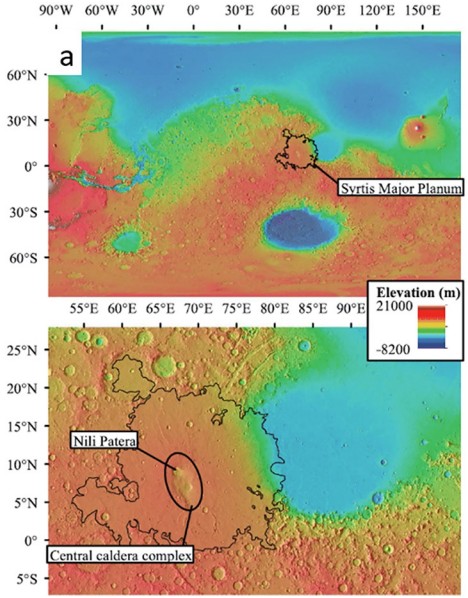

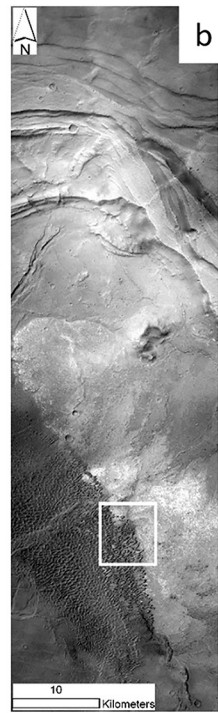

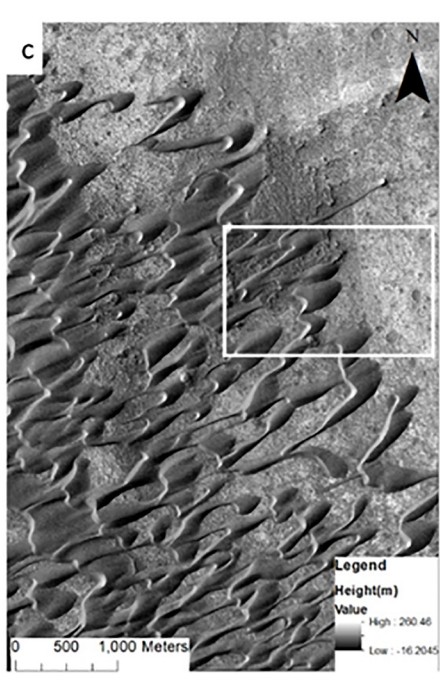

**Fig 3. Showing the location of the study site.** (a) *Top*: MOLA Topography map showing the location of Syrtis Major Planum. *Bottom*: Detailed view of the caldera complex, with the ellipse surrounding the northern caldera of Nili Patera and the southern caldera of Meroe Patera. This image is extracted from Fawdon et al., (2015) [67]. (b) CTX image of the barchan dune field located within the south-east part of the Nili Patera caldera, the white box indicates the location of image c. (c) The white box contains the portion of the dune field containing the dunes of interest for the CFD simulations.

eastern edge of the caldera which provides a more topographically uniform terrain to simulate wind flow over using CFD. The dune field is symmetric with barchan dunes 200–400 m long [66], with dune slipfaces facing the southwest.

Barchan dunes are formed in areas with unidirectional wind flow, and a limited sand supply [68–70]. A barchan dune typically has a crescentic shape, with a shallow angled windward slope (typically 10˚–14˚ on Earth [71], and a downwind lee slope which is at the angle of repose [72]. When wind flow is perpendicular to the dune crest, flow separation and recirculation occurs between the two 'horns' of the dune (Fig 4). When large enough, aeolian features can modify and distort the direction of the wind flow and cause substantial acceleration/deceleration of the velocity profile [73]. The large barchan dunes with an average height of 35 m in our

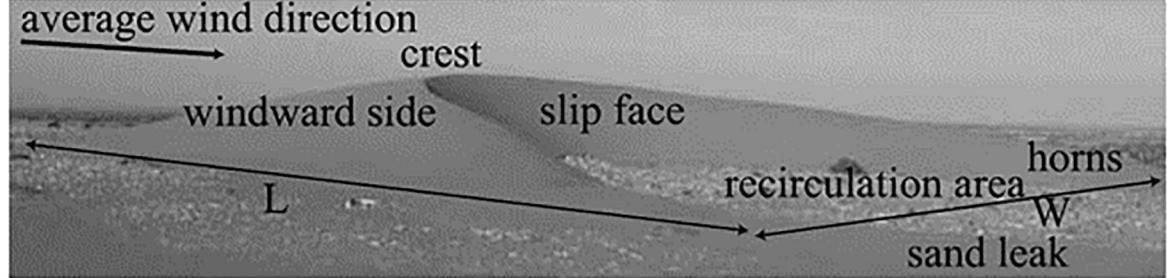

**Fig 4. Diagram showing the typical shape of a barchan dune, the Length (L) and Width (W) shown, as well as the unidirectional wind flow which causes the uniform shape in all barchan dunes.** Image from Hersen (2005) [73].

study site shown in Fig 3(c) allow for the examination of the relationship between large aeolian features and the wind flow velocity outlined [73].

A substantial amount of research has been performed at the Nili Patera dune field [2,17,18,74]. The dune field has been extensively monitored and imaged by HiRISE. Temporal HiRISE images of the site were examined in order to determine levels of sand transport in the region [18], using the Co-registration of Optically Sensed Images and Correlation (COSI-Corr) process [75]. The resolution of the HiRISE camera limits the aerial extent of the imagery, so coarser Context Camera (CTX) imagery has previously been used to calculate dune migration rates on a longer temporal basis in this area [66]. Sediment transport in Nili Patera has been observed from both HiRISE and CTX imagery, despite the previously mentioned paradox between sediment transport and observed transport capable winds. All these factors make the site of interest for examining CFD microscale wind dynamics within the scope of the proposed methodology.

## Computational Fluid Dynamics (CFD)

Microscale wind flow simulations were performed utilising an open-source C++ library for Computational Fluid Dynamic (CFD) simulations, OpenFOAM [76–78]. CFD is a numerical method of performing simulations in which the governing differential equations of fluid flow are solved to obtain a numerical description of the flow field under investigation [79]. The Navier-Stokes equations account for the governing equations of fluid flow including the conservation of mass, conservation of momentum and conservation of energy [80,81].

The solver PIMPLE was used in this study, which is based on an algorithm that combines the solvers 'Pressure-Implicit with Splitting of Operator' (PISO) and 'Semi-Implicit Method for Pressure Linked Equations' (SIMPLE). PIMPLE was used as the flow solver in this study due to its large-timestep transient nature. The turbulence model employed for this simulation was the RNG (Re-normalization group) κ-ε model, which is used to renormalize the Navier stokes equations, accounting for the different scales of motion, particularly small-scale motion and rotating flows. This model calculates turbulent kinetic energy (κ) and dissipation rate of turbulent kinetic energy (ε) [82]. Using the RNG κ-ε model enables the simulation to calculate the mean flow distortion turbulence and yields more realistic microscale wind flows over complex topography. The RNG κ-ε model was selected due to its proven performance in separated flows which occur downwind of many aeolian features [81,83–88], as well as the ability to accurately account for the effects of swirling flows [89]. The selection of this model for this study was also appropriate as it is capable of yielding more accurate results regarding recirculating flow and improved prediction of separated flow in comparison to the original k-epsilon model [90].

In this application, when the velocity residual values were 5 orders of magnitude smaller than the maximum residual, the solution was deemed to have reached a sufficient level of convergence. For the topographic surface, 0.25-m resolution HiRISE imagery of the site was used to produce a 1-m resolution DTM. The DTM was created from stereo image pair ESP_017762_1890 and ESP_018039_1890, acquired 22 Earth-days apart. The DTM was converted into a Stereolithography (STL) file in order to be incorporated into the computational domain for CFD simulations. The domain measured 2150 m x 1070 m x 475 m. The vertical domain height of 475 m was specified to ensure that the maximum vertical extent of the domain was at least 5 times the height of the tallest dune in the study site [15,85,91]. A simulation with a 10 km vertical extent was also simulated to compare against the 475 m vertical extent case, and no significant difference in results was observed.

Wind speed and direction were extracted from the MRAMS output in order to include winds, possibly modified by the regional km-scale topography, which would be responsible for

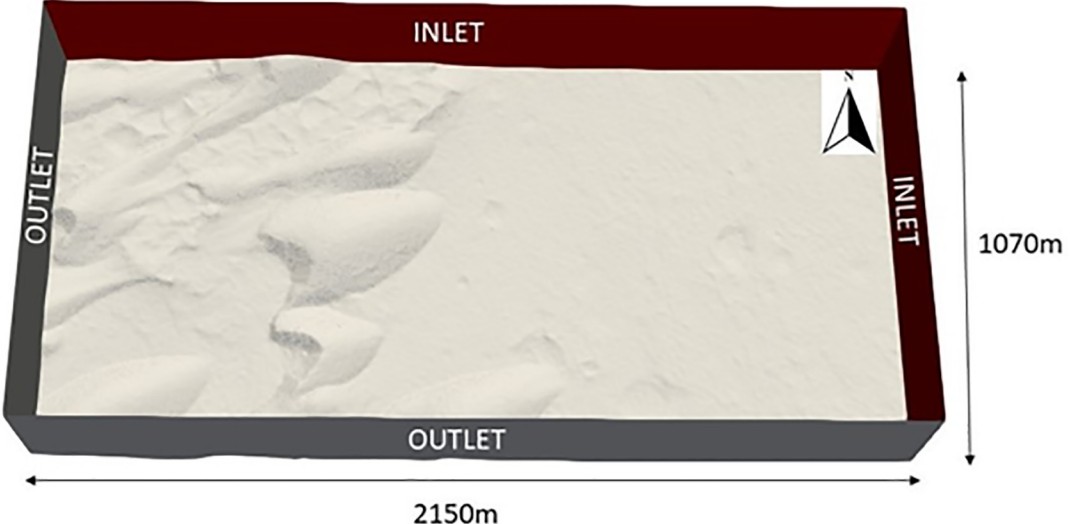

**Fig 5. Showing the CFD domain boundary setup.**

the modification of the Nili Patera barchan dunes. Winds at a single MRAMS point just upwind of the dune field were examined every 20 Mars-minutes for 3.5 simulated sols (at a season of $L_s$ ~270˚). The samples that had a friction velocity which exceeded the impact threshold for saltation on Mars were considered winds which could be responsible for the modification of the dune field, and when averaged, resulted in an incident wind direction of 67.67˚ at 11.65 m s$^{-1}$, 5 m above the surface. These wind conditions were implemented as the horizontally uniform initial boundary conditions for the CFD simulation setup. The vertical boundary conditions for the simulations used individual vertical height wind data from MRAMS output data, with 17 vertical MRAMS layers being used in order to implement a non-uniform wind velocity boundary layer. The boundaries of the computational domain and the STL of the study site are seen in Fig 5.

Turbulent kinetic energy (k) and turbulence dissipation rate (ε) were calculated using Eqs 1 and 2, assuming a constant friction velocity ($u_*$) defined by [92]:

$$k\left(z\right) = \ \frac{u_*^2}{\sqrt{C\mu}} \tag{1}$$

$$\varepsilon(z) = \ \frac{u_*^3}{k(z + z_0)} \tag{2}$$

where $C\mu$ is a constant of the κ-ε model equal to 0.09 [92], $u_*$ was set to 0.3 m s$^{-1}$, κ(z) and ε(z) where z is the height of interest, κ is the von Karman constant equal to 0.42 and $z_0$ is the physical roughness height equal to 0.01 m.

The computational domain for the CFD simulation contained approximately 27 million cells, decreasing in size from the top of the domain to the surface. At the surface of the computational domain, the vertical and horizontal cell sizes were 1.9 m and 1.67 m respectively. As the simulation velocity residuals had converged, the simulation was run for 235 wall time seconds in total. An inlet speed of 11.65 m s$^{-1}$ took 185 seconds to traverse the 2150 m long domain. The simulation was run for a further 50 seconds to examine the CFD output. The additional 50 seconds was included to ensure there was no significant difference in wind speed

results between the time taken for the wind flow to traverse the domain (the first 185 seconds) and in the following 50 seconds.

The CFD simulation performed in this study has not accounted for the 'terra-incognita' zone, also known as the gray zone [93–95]. In this region, the full scale of the planetary boundary layer eddies are no longer smaller than the size of the mesoscale model mesh, as a result of improvements in computational power [93,96]. When mesoscale models are run at such high resolutions, they can begin to 'double count' eddy effects [23,97] as a result of imperfect turbulence parameterizations, which then may influence results of the mesoscale modelling output which has a knock-on effect to the CFD microscale modelling. However, the degree to which this region affects results is unknown due to the lack of boundary layer measurements on Mars [97]. During the day on most of Mars, the scales of dominant eddy motion in the planetary boundary layer range from 2 to 10 km or more. Properly accounting for the transition within this region requires a comprehensive down-scaling model which uses different and appropriate turbulence parameterizations for each grid resolution which was beyond the scope of the present investigation. Further limitations to the CFD simulation used in this study include the use of a horizontally homogenous initial wind field, this is a result of the ~2 km horizontal grid spacing in the MRAMS simulation, while the East-West inlet boundary of the simulation was 2150 m. In this study, only one upwind spatial grid point was selected for the initial CFD conditions as we saw little observable difference between adjacent grid points upwind of the selected region of interest shown in Fig 3c (unlike elsewhere in the caldera), and our present study does not require multiple or complex lateral inlets to the CFD domain.

Examining the output of the MRAMS simulations shows that there is a pattern of diurnal variation in wind speed and direction over the study site. As previously outlined, this study extracted the wind conditions associated with potential sediment transport from MRAMS near the CFD domain inlet and averaged those specific winds. In this way we account for these diurnal variations and provide a broadly representative wind condition for the study site for the martian season being studied. During the season examined, 73 of the 254 total time steps of the MRAMS output were associated with potential sediment transport—performing CFD simulations for each of the 73 individual time steps is not possible due to computational limitations. Averaging the 73 values to provide representative values of wind conditions at this season for 17 vertical layers of the domain is an appropriate method (given the singular CFD inlet conditions needed) to account for some of the varying diurnal atmospheric behaviour. The average incident wind direction from each of the time steps provided a realistic overall representative condition of the total winds at this season (Fig 6).

## Mesh independence study

A mesh independence check was conducted for this study to ensure that the results of the simulation were independent of the mesh size (Fig 7a). Wind speeds 1.9 m above the surface of the STL were taken over 20 points along the profile of a large barchan dune at the study site. Successively finer meshes, refining the mesh by a constant factor of 1.5 (versus the preceding mesh) and the same input parameters were used until the results converged. Any discrepancy between values from the 1.67 m and 1.1 m horizontal mesh-size test simulations were less than 1.5%, so the mesh was considered independent of the results. Therefore, the simulation employed a 1.67 m horizontal mesh size (Fig 7b). The cross-section of the barchan displayed in the graphs of Fig 7(a) and 7(b) is seen in Fig 8.

## Barchan dune flow dynamics

Changes in near surface wind over a barchan dune are well documented [86,98–101]. On the windward side, wind flow decelerates before the toe of the dune, accelerates rapidly until the

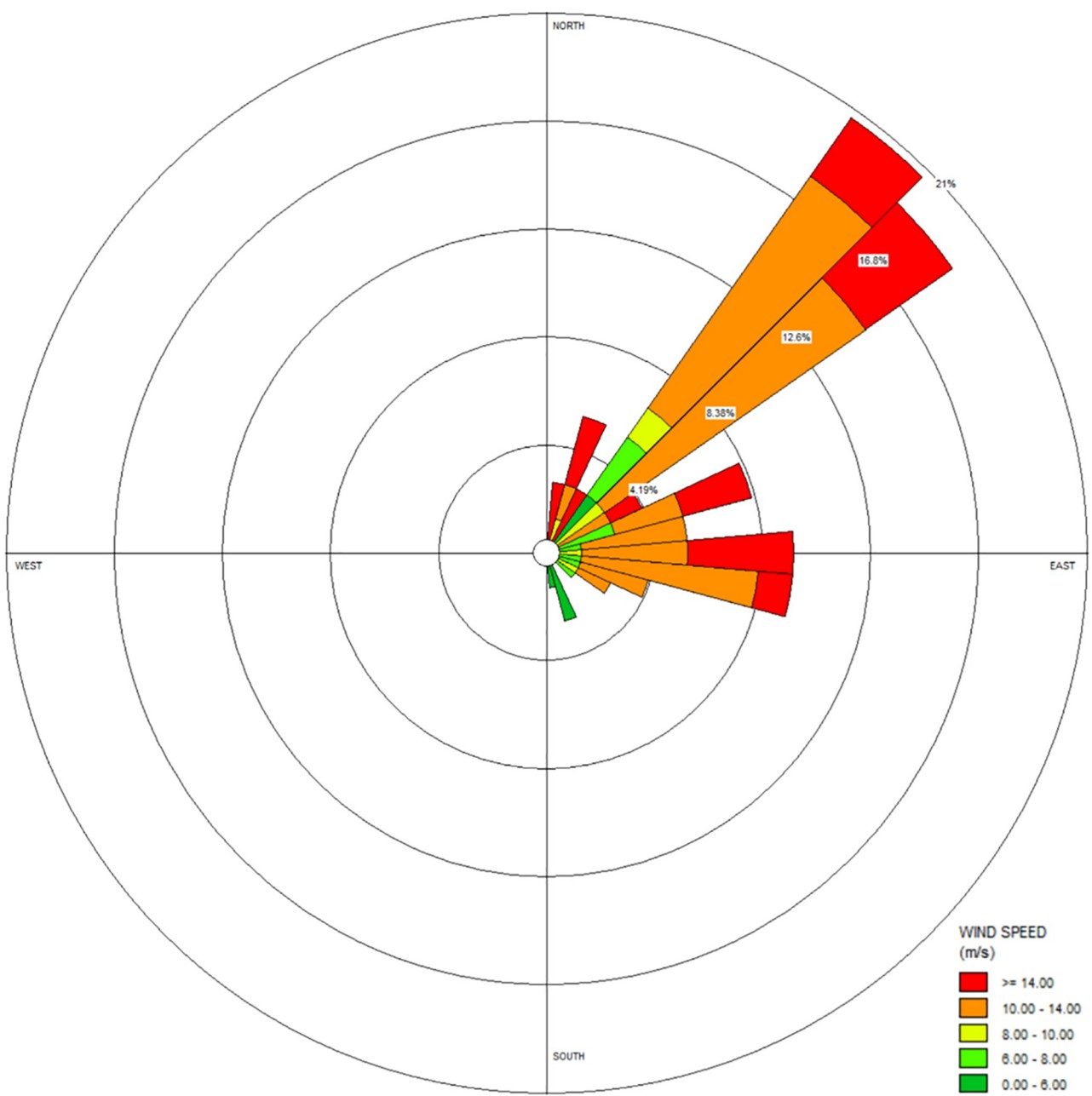

**Fig 6. Wind rose of the 73 winds which exceed the friction velocity for sediment transport, which gave an average incident direction of 66.4.**

crest of the dune where it reaches its maximum, followed by rapid deceleration in the lee of the dune, with a separation bubble of recirculation [73,86,98,102–104] before the flow settles again further downwind of the barchan (Fig 9).

Output from the CFD simulation was analysed to examine the ability of our set-up to reproduce the conditions of wind flow over a barchan dune field as described in [73,104]. A well-reproduced CFD profile of wind flow over a dune field is a necessary prerequisite to accurately predict local sediment flux [102]. As described in the following section, the CFD results from this study replicate patterns of near surface wind as described by previous flow dynamics

(A)

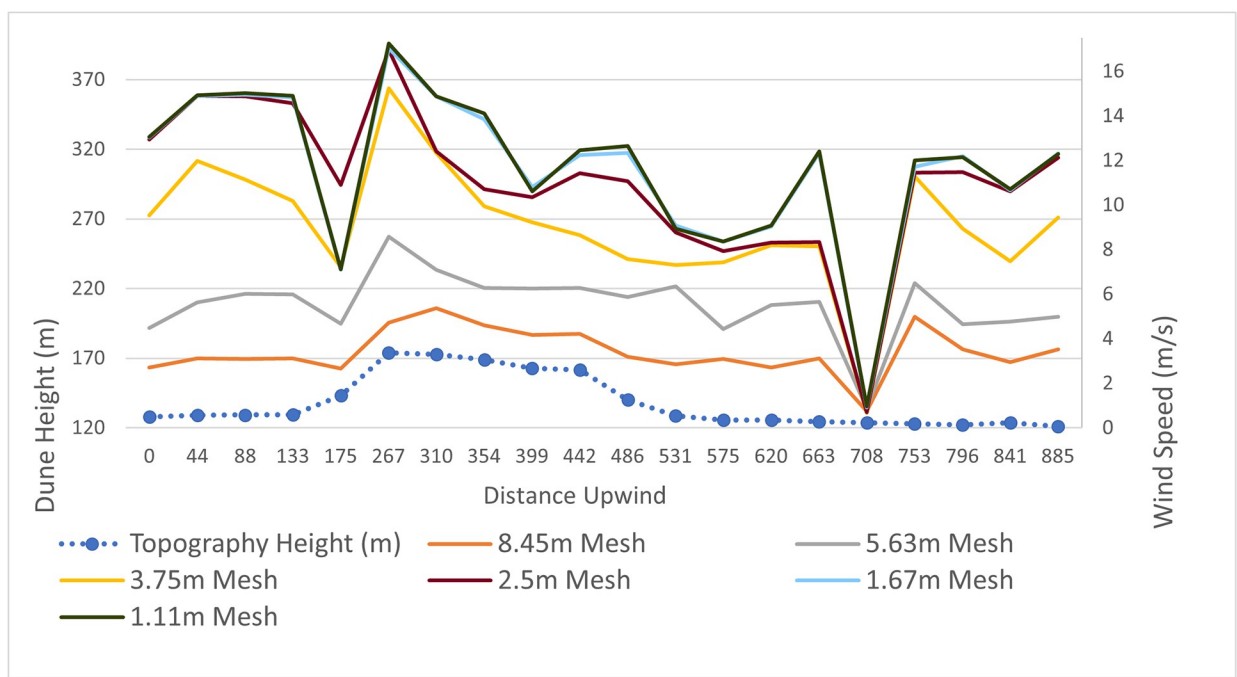

(B)

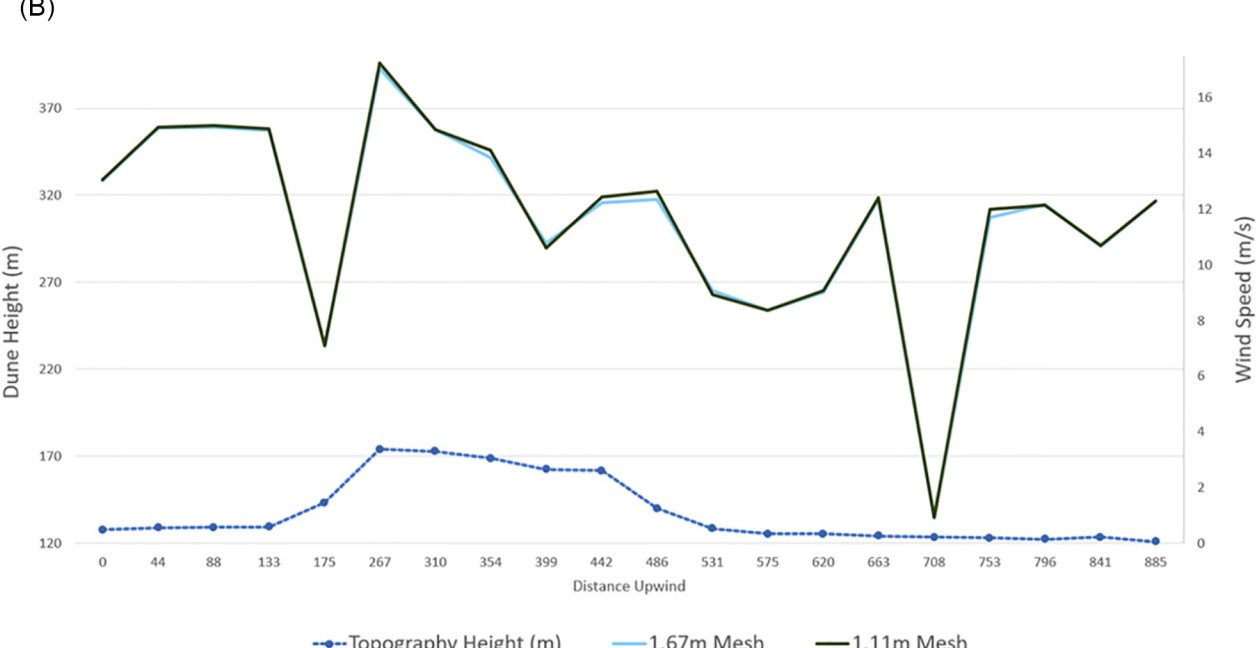

**Fig 7. a**: Wind speeds 1.9 m above the surface over a barchan dune at the study site, from coarse and fine mesh resolutions. Results converged at a 1.67 m mesh resolution; the 1.67 mesh line is behind the 1.11 m mesh line. **b**: Wind speeds for the 1.67 m and 1.11 m meshes, 1.9 m above the surface of the dune.

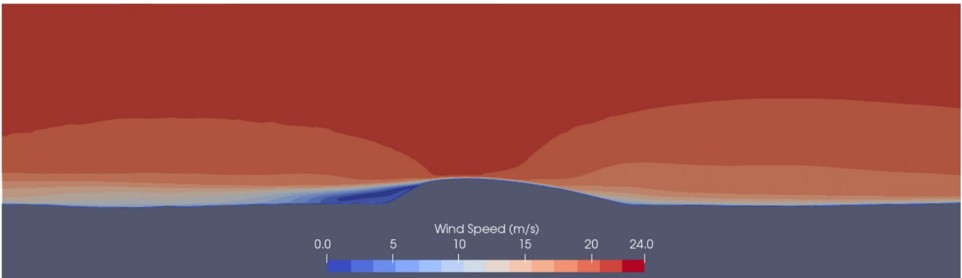

**Fig 8. Wind profile over the central barchan shown in the mesh independence check of Fig 6.**

studies over barchans, as well as previous research examining wind flow speeds and processes in the martian environment.

## Results

### Wind speed

The CFD simulation performed in this study reproduced the wind flow pattern over barchan dunes (as depicted in Fig 9) well. The wind speed map (Fig 10) shows that the CFD simulation reproduces the flow dynamics of a barchan dune outlined [102]. The wind speed upwind of the barchan field is consistent at 12–14 m s$^{-1}$, which decelerates to less than 3 m s$^{-1}$ at the toe of each barchan dune at the study site. The wind flow over each of the barchan dunes then undergoes rapid flow acceleration up the stoss slope of the dune before reaching the maximum wind flow speed at the crest of the dune—each dune at the study site had an individual maximum wind flow speed, ranging from ~20 m s$^{-1}$ to ~29.5 m s$^{-1}$. After reaching individual maximum wind flow speed at the crest of each dune, the flow rapidly decelerates in the lee of each barchan dune to <1 m s$^{-1}$.

The wind flow dynamics of Dune 1 (Fig 10) are shown in Fig 11. The high wind speeds calculated in Fig 11 (~10 m above the surface of the dune) were not ubiquitous across the dune field. As seen in Fig 12, for the smaller barchan dune (Dune 2 Fig 10), the maximum speed at the same height above the dune crest is 22 m s$^{-1}$. The maximum wind speeds of 27.5 m s$^{-1}$ calculated in this study's CFD simulation (Fig 11) are similar to the lower boundary of the free stream velocity of ~24–27 m s$^{-1}$ 10 km above the surface calculated by a GCM [2]. Wind speeds greater than 20 m s$^{-1}$ are thought to be more typical of dust storms on Mars, rather than average conditions [56]. MRAMS and MarsWRF mesoscale modelling was performed in Jezero Crater (18.38˚N 77.58˚E) located in Syrtis Major Quadrangle [97]. Up-slope wind speeds with a maxima of 16 m s$^{-1}$ at two seasons (Ls = 0˚ and Ls = 180˚) were calculated [97], these modestly high wind speeds were calculated by both of the mesoscale models used in that study. The discrepancies between the wind speeds calculated by the GCM and mesoscale modeling, and

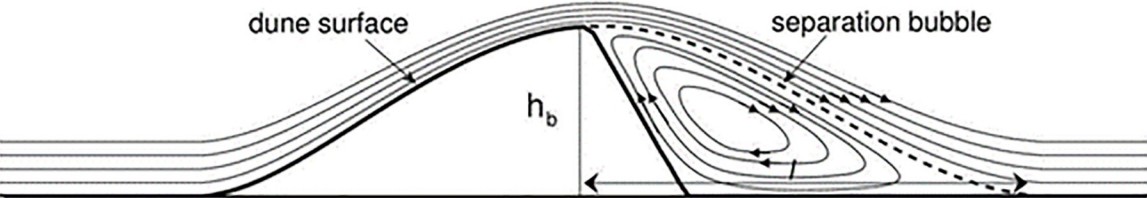

**Fig 9. Diagram adapted from Durán et al. (2010) [104] showing the separation bubble of recirculating flow in the lee of the dune.**

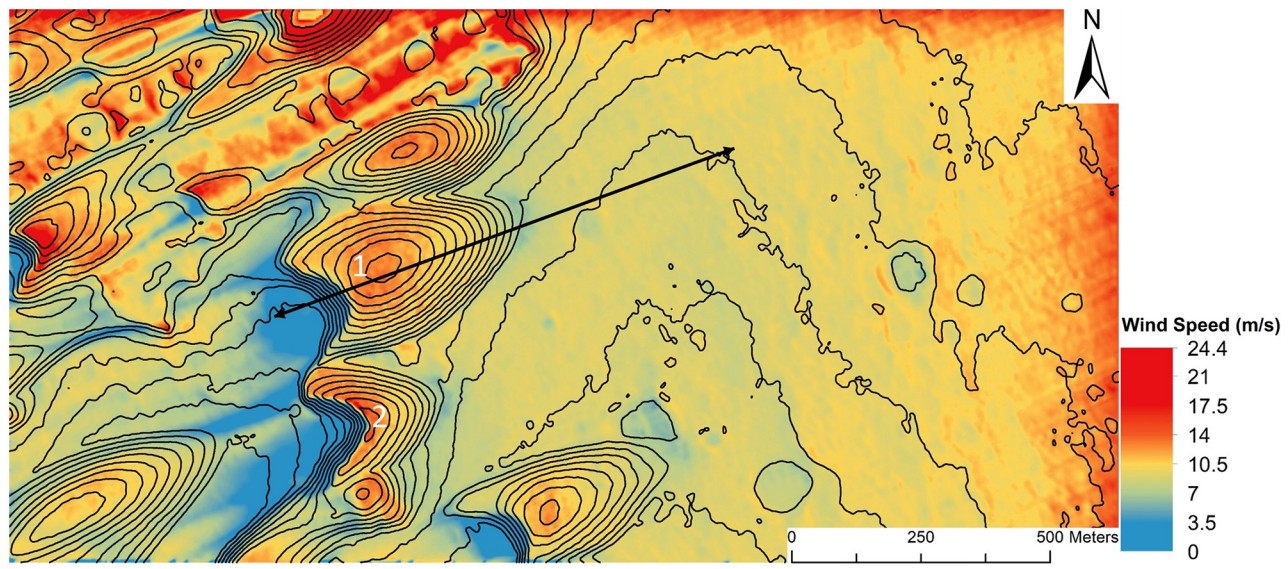

**Fig 10. Showing time-averaged wind speeds 1.9 m above the surface of the STL over the study site in Nili Patera from the CFD simulation.** Inlet wind direction flows from 66˚ angle from upper right to lower left, shown in the wind streaks. The arrow across the central barchan shows the orientation and length of the domain probed for the mesh independence check (Fig 7a and 7b).

the wind speeds calculated over the study site from the CFD simulations may be simply explained by the ability of the microscale CFD simulations to better account for the effect of sub-dune length-scale topography. The very high maximum spatial resolution of the CFD (1.67 m) is over several orders of magnitude greater than the resolution of either the GCM or mesoscale models which calculated the wind speeds above. This means that the CFD simulation is able to account for and resolve wind flow behaviors such as the very localised topographic acceleration of wind flow which larger-scale models are unable to resolve. Thus, in this study, the microscale CFD simulation's ability to account for this effect helps explain the higher wind flow speeds, particularly at the crest of large dunes.

The flow dynamics of Dune 2 (Fig 10) are further shown in Fig 13. The rapidly decelerated flow in the lee of the dune did not reach speeds observed upwind of the initial obstacle until the flow processes over the upwind dunes was repeated (Fig 10). That is, wind flow that decelerated in the lee of the dune required an obstacle such as a barchan dune further downwind to re-accelerate the wind flow before the wind was capable of reaching the speeds observed upwind of the dune field. In Fig 10, the main body of wind flow over the large central barchan (Dune 1) reaches a maximum speed of ~27.5 m s$^{-1}$, (Fig 11) but in the lee of the dune, the majority of the wind flow is decelerated to <3 m s$^{-1}$. This wind flow remains at this low speed for ~200 m downwind of the main barchan before it reaches the southwestern barchan at the outlet boundary, which repeats the described flow dynamics over a barchan dune, inducing topographic acceleration and reaches wind speeds observed in the upwind of the eastern edge of the domain again. Each of the barchans in the study site show the described pattern of decelerated flow at the toe of the windward dune slope, followed by acceleration until the crest of the dune and rapid deceleration to very low wind speed in the lee of the dune. Similarly, each of the barchans experienced recirculating flow in the lee of the dune, similar to the pattern seen in Figs 12 and 13.

Larger-scale modeling output provides a good initial comparison to the wind speeds calculated by the CFD simulation in this study. However, for a more precise verification of results, the wind speeds calculated in this study must be compared to contemporary microscale/*in situ*

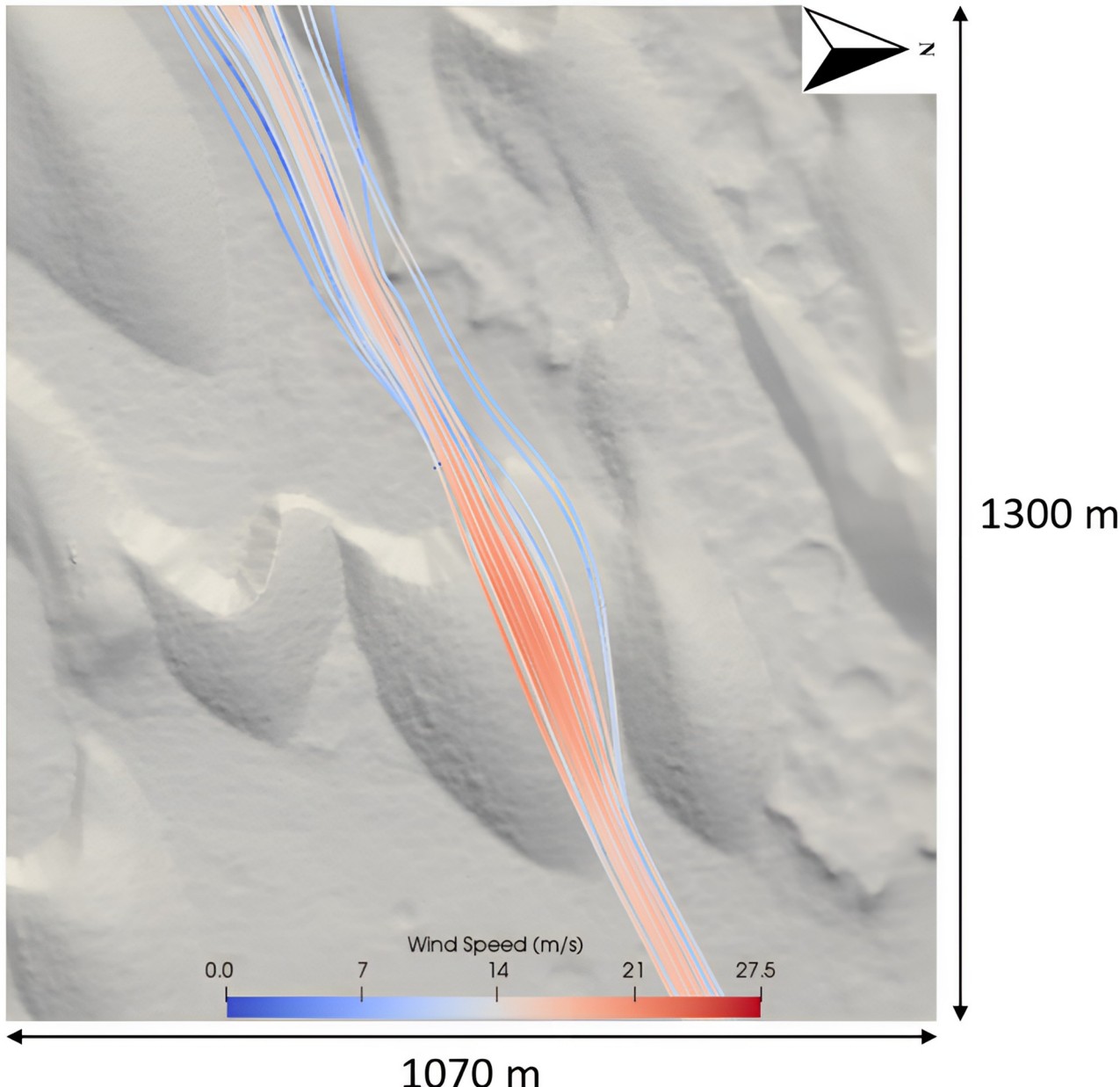

**Fig 11. Wind flow over the large central barchan (Dune 1 in Fig 10).** The morphology of the dune field distorts the speed and direction of the wind flow.

studies examining wind speeds on Mars. Wind speeds greater than the 20 m s$^{-1}$ calibration of the Rover Environmental Monitoring Station (REMS) onboard the curiosity rover in Gale Crater (5.4˚S 137.8˚E) have been observed [105], however wind speeds >10 m s$^{-1}$ are the most commonly observed wind flow in Gale Crater. Data collected by REMS have been used as the initial input conditions for CFD simulations [4]. In spring and autumn, REMS recorded wind speeds as high as 15 m s$^{-1}$ and 17 m s$^{-1}$, respectively (Mars Year 33). These *in situ* values are several meters per second greater than the average upwind conditions in this study's CFD simulation, but the maximum stoss slope speed calculated in this simulation is ~10 m s$^{-1}$ higher

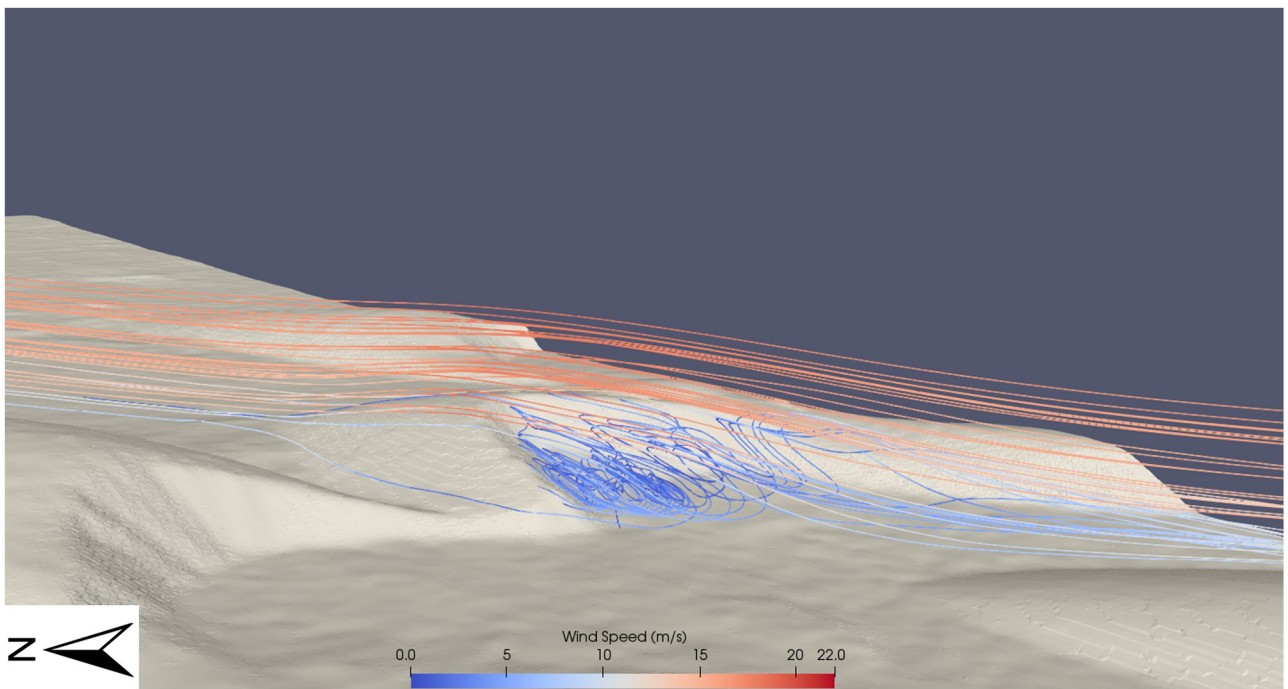

**Fig 12. Showing recirculation of wind flow in the lee slope of a barchan dune.** Wind flow is from left to right in this image. The wind accelerates to a high of 22 m s$^{-1}$ at the crest of the dune, before rapidly decelerating in the lee of the slope.

than the maximum observations by REMS. Unfortunately, due to the lack of surface coverage from Mars landers, it is difficult to make a direct comparison between the *in situ* data collected from REMS (which has not been on the crest of a barchan dune) and calculated CFD wind speed value at our study site, as the local wind regimes differ vastly between locations on Mars. Despite this, the upwind speeds in this study's simulation are similar to the wind speeds collected from REMS, which provides broad confidence that the CFD simulation flow over the upwind terrain is plausible and that the much higher dune crest wind speeds come from fully accounting for the effect of the large barchans accelerating the flow to much higher speeds.

Wind flow behaviour similar to the CFD output from this study were observed in CFD simulations for Proctor crater (Jackson et al., 2015). The wind flow simulations over the dunes within Proctor crater showed similar patterns of significant upslope acceleration over individual dunes in this study seen in Fig 11. The topographic acceleration over dune crests in a martian environment was also previously observed [56], they performed CFD simulations which closely matched the pattern of topographic features in Arnus Vallis. A future method to verify the CFD output data using this methodological approach would be to test the whole technique at a site which has had meteorological data returned from a successful lander. However, the purpose of this study was to outline an approach which would allow microscale wind conditions at sites on Mars which did not have reliable *in situ* meteorological data to be studied using a combination of modeling scales.

## Wind flow steering

Barchan dunes are capable of distorting the localised surface wind flow [73,101]. The flow dynamics over the large central barchan (Barchan 2 in Fig 10) are shown in Fig 11. This barchan is 54 m tall from the toe of the dune to its crest. The streamline speeds vary with height

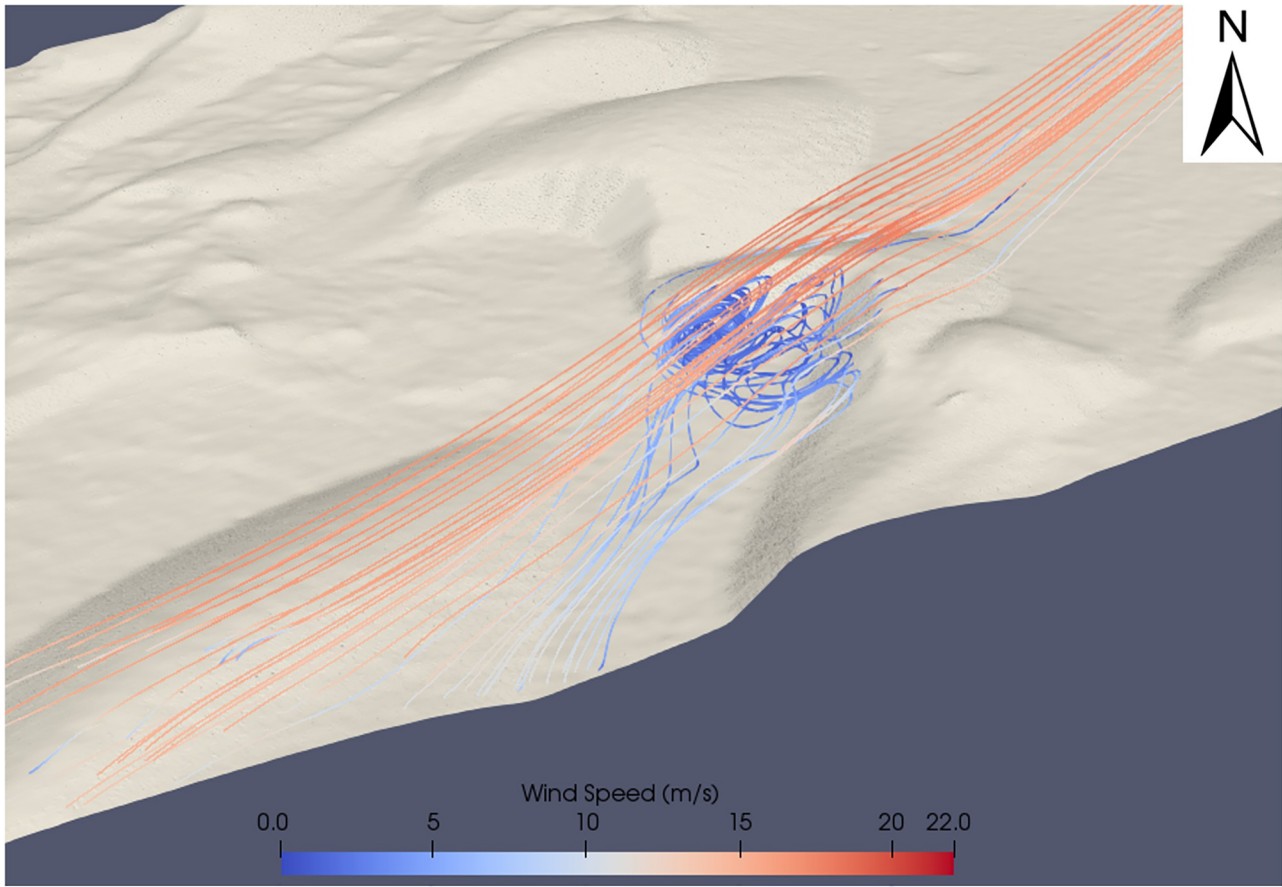

**Fig 13. Showing the reverse image of the recirculating flow shown in Fig 10.** Further to the recirculating flow, there is also additional downwind distortion of wind flow which does not rejoin the main body of wind flow.

across the surface of the dune and therefore the deceleration observed at the toe of the dune is not as well observed in this aerial view of the dune. Fig 11 highlights the effect that the barchan shape can have on distorting the wind flow. When the uniform body of wind flow in Fig 11 reaches the toe of the dune, it is distorted and the body of flow splits, with a portion of the wind flow steered around the northern face of the dune rather than over the top, and the wind flow is then distorted further by the northern face of the dune in the southwest of the image. The flow dynamics of dune 2 in Fig 10 are shown in Figs 12 and 13. These figures highlight the previously described pattern of flow acceleration over the crest of the dune and clearly shows the rapid deceleration and recirculation of wind flow above the lee slope of the barchan dune. There is a large area of recirculating flow over the lee slope, at speeds as low as 0.5 m s$^{-1}$, before the flow reattaches to the main body of flow further downwind. The right edge of the image is ~150 m downwind of the zone of recirculation, and it is evident that the near-surface boundary layer had not fully recovered to the pattern observed upwind of the study site.

The upslope acceleration until a maximum speed at the crest of dunes (Figs 12 and 13) and rapid deceleration in the lee of the dune within a zone of recirculation corresponds well with wind flow behaviour over barchan dunes [73,104]. Fig 12 highlights the previous statement that the wind flow profile of (Fig 9) [104] is well reproduced. Both images show flow compression on the stoss slope, followed by a significant region of recirculation in the lee of the dune, and Fig 12 also shows the highlighted separation bubble between the horns of the dune seen in

Fig 9. The complex wind flow pattern in the lee of the dunes is similarly seen in Fig 13 (Dune 2 of Fig 10). The reverse view of the dune shows both the recirculation between the horns of the dunes, but also the significant flow distortion in this region, which does not rejoin the main body of flow further downwind. The modeling results presented in this study show that the CFD set-up in this study is capable of well reproducing the described spatial pattern of wind flow over the barchan dune field, providing insights into the steering, separation and changes in wind flow speed over individual dunes at the study site. This insight provides an understanding of the flow structures over the dune field which cannot be resolved by larger-scale modelling, such as flow separation downwind of the lee slope, showing that this approach can be applied to sediment transport studies in order to further the understanding of aeolian landform dynamics, particularly at sites that lack *in situ* data.

## Conclusions

This study presents a procedure for creating a simulation of wind flow on Mars using larger-scale atmospheric modeling to inform a subsequent CFD microscale (meter-scale) simulation. This method uses output from a km-scale mesoscale simulation of regional-to-local atmospheric conditions to construct the driving conditions for very high resolution (<2 m resolution) CFD simulations over similarly-high-resolution topography derived from orbital images of the surface. Simulating such microscale wind flow allows for the investigation of process-response relationships over small-scale topography. In lieu of high quality *in situ* data at every location of interest on the planet, our process has yielded results which show that this methodology can reproduce high-resolution flow dynamics over a barchan dune on Mars: Flow deceleration at the dune toe, acceleration to a wind speed maximum at the crest of the dune, and sharp deceleration into an area of re-circulated flow in the form of a separation bubble at the lee slope of the dune (observed at each of the barchan dunes in the study site).

By using realistic kilometer-scale mesoscale atmospheric modeling to assess the wind field just upwind of the target study site at a given season, the resulting CFD microscale simulations are informed with realistic atmospheric conditions to examine wind flow patterns over the study site. This work makes certain assumptions, including neglecting the details of the turbulent flows within the terra incognita zone, in order to explore this novel methodological approach to examine CFD microscale winds across a dune field in a computationally practical manner. Despite this, the general conclusions of this study have highlighted the feasibility of using a combined-scale methodology to examine microscale near-surface wind flow on Mars, using a portion of the Nili Patera dune fields as a test site to produce sub-dune length-scale output of wind flow conditions over a barchan dune field on Mars. This technique is applicable elsewhere on Mars where sufficient HiRISE DTM information is available. Future studies utilising this methodology may consider using time-variant boundary conditions and more-detailed microscale atmospheric processes in order to further improve the realism of the CFD microscale simulations.

## Author Contributions

**Conceptualization:** Derek W. T. Jackson, Timothy Michaels, Jean-Philippe Avouac, Andrew Cooper.

**Investigation:** Richard Love, Derek W. T. Jackson.

**Methodology:** Richard Love, Timothy Michaels.

**Project administration:** Derek W. T. Jackson.

**Resources:** Jean-Philippe Avouac.

**Software:** Richard Love, Timothy Michaels, Thomas A. G. Smyth.

**Supervision:** Derek W. T. Jackson, Thomas A. G. Smyth, Andrew Cooper.

**Validation:** Richard Love.

**Visualization:** Richard Love.

**Writing – original draft:** Richard Love.

**Writing – review & editing:** Derek W. T. Jackson, Thomas A. G. Smyth.

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
