## [Decision Letter · Decision Letter 0]

13 Sep 2022

PONE-D-22-22872Full Title: From Macro- to Microscale: A combined modelling approach for near-surface wind flow on Mars at sub-dune length-scalesPLOS ONE

Dear Dr. Love,

Thank you for submitting your manuscript to PLOS ONE. After careful consideration, we feel that it has merit but does not fully meet PLOS ONE’s publication criteria as it currently stands. Therefore, we invite you to submit a revised version of the manuscript that addresses the points raised during the review process.

We look forward to receiving your revised manuscript.

Kind regards,

Omer San

Academic Editor

PLOS ONE

3. We note that Figures 2a, 2b ,2c and 9 in your submission contain [map/satellite] images which may be copyrighted. All PLOS content is published under the Creative Commons Attribution License (CC BY 4.0), which means that the manuscript, images, and Supporting Information files will be freely available online, and any third party is permitted to access, download, copy, distribute, and use these materials in any way, even commercially, with proper attribution. For these reasons, we cannot publish previously copyrighted maps or satellite images created using proprietary data, such as Google software (Google Maps, Street View, and Earth). For more information, see our copyright guidelines: http://journals.plos.org/plosone/s/licenses-and-copyright.

a. You may seek permission from the original copyright holder of Figures 2a, 2b ,2c and 9 to publish the content specifically under the CC BY 4.0 license. 

Additional Editor Comments :

Reviewers have now commented on your paper. You will see that they are advising that you revise your manuscript. If you are prepared to undertake the work required, I would be pleased to see your revised manuscript and reconsider my decision. Therefore, if you decide to revise the work, please submit a list of changes or a rebuttal against each point which is being raised when you submit the revised manuscript.

Reviewers' comments:

Reviewer's Responses to Questions

**Comments to the Author**

1. Is the manuscript technically sound, and do the data support the conclusions?

Reviewer #1: Partly

Reviewer #2: Yes

2. Has the statistical analysis been performed appropriately and rigorously? 

Reviewer #1: No

Reviewer #2: Yes

3. Have the authors made all data underlying the findings in their manuscript fully available?

Reviewer #1: No

Reviewer #2: Yes

4. Is the manuscript presented in an intelligible fashion and written in standard English?

Reviewer #1: Yes

Reviewer #2: Yes

5. Review Comments to the Author

Reviewer #1: The paper covers an interesting topic using a valid modeling methodology. However, the manuscript lacks details that are necessary to make the work reproducible. The quality of figures and associated descriptions needs substantial improvements. More review comments are included in the attached report.

Reviewer #2: The paper presents a very interesting study on wind flows over barchan dunes for wind flow regimes that can sustain sediment transport over barchans but excluding the "terra incognita" region corresponding to the flow on the lee-side of the barchan and within the horn region. The study is a numerical study coupling macro and microscale models. The paper is generally well written, and the results presented are robust and interesting. I recommend the paper for publication after some minor revisions.

General Comments:

1) There is no reason why you would not be able to model the terra incognita region with OpenFoam, so I am not sure why this region was excluded. Would you need more computational power so that the mesh size is sufficiently small to reproduce the complex turbulence on the lee side of the Barchan?

2) Are you coupling the models offline or using the ESMF? I would have thought you could use ESMF for this problem and it possibly would make the workflow more straightforward.

Specific comments:

line 362: I would say "Turbulent kinetic energy" not "Turbulence kinetic energy"

line 370-371: "is set" not "is equivalent"

line 371: replace "k(z) and epsilon(z) are the wind speeds at height z" with "z is the height of interest"

line 378: replace "simulation was simulated" with "simulation was run"

line 381: "and in the time following" is not clear, please revise.

line 393: replace "more, properly" with "more. Properly"

line 395: replace "beyond what our present investigation required" with "beyond the scope of the present investigation"

line 400: "immediate locale" not clear, please revise.

line 408. "73 of the 254 total time steps" are we talking about time steps, of about "wind states" as line 410 suggests? if you are seeing diurnal patterns, I guess your simulations are run for at least a couple of days but I am not sure I saw the period over which you are running the codes in the paper.

line 552: "on Mars" not "on the Mars"

line 604: remove "more-informed"

6. PLOS authors have the option to publish the peer review history of their article (what does this mean?). If published, this will include your full peer review and any attached files.

Reviewer #1: No

Reviewer #2: **Yes: **Dr. Vanesa Magar

---

## [Author Response · Author response to Decision Letter 0]

4 Oct 2022

Review of :

“From Macro- to Microscale: A combined modelling approach for near-surface wind flow on Mars at sub-dune length-scales”

The authors present a nested-modeling approach based on a hierarchy of multi-scale solvers to analyze the near-surface wind flow on Mars. A macro-scale general circulation model feeds a meso-scale model which in turn is used to define the boundary conditions for the micro-scale CFD solver. The idea of linking multi-scale models to progress from large scale phenomena to finer scale processes is well-established and the application to study the wind flow on Martian terrain is very interesting. Nonetheless, in its current form, the paper is hard to follow and most of the algorithmic development details are missing. Below, I list a few comments that the authors might want to consider.

Please note, all line references correspond to the ‘clean’ manuscript

1. Pages 9 to 20 were assumed to provide the reader with an understanding of the employed models from NASA-AMES GCM to MRAMS, and finally to the OpenFOAM solver. Instead of that, a very broad and generic description of these models is presented. The reviewer suggests providing more details on the computational setup and a formal definition of the mappings between different levels of solvers.

In regard to the GCM, at line 209, we direct authors to ‘Haberle et al., 2019’, where the model is described in great detail. At line 217 we have also included that the GCM uses a ‘a simple cylindrical map projection for the horizontal grid.’

The description of mapping from GCM to MRAMS was described, however this has now been clarified. 

Line 224-228: ‘Details of this type of pre-processing, in order to successfully implement GCM data as the initial state for MRAMS simulations can be found in published work, such as Rafkin et al. (2001) and Rafkin and Michaels (2019). Similarly, Toigo and Richardson (2002) describe the implementation of GCM data for use in a mesoscale model.’

The mapping of MRAMS to CFD, including the specific details of filtering of data are included throughout the text, starting at line 398.

The CFD in particular is described in detail starting at line 360, outlining the specific solver, the turbulence model, height of the domain and equations used for it, with references for each description. The details of the DTM are listed and the domain specifications. We feel that the CFD solver is outlined in sufficient detail and we have provided a comprehensive background for researchers attempting to repeat this method.

2. Relevant to my previous comment, the authors refer to some terminologies without prior introduction of those. For example, in Page 12 - Line 257, the authors refer to Grid 1 and Grid 5, but those were not defined. It would be helpful to clearly state the type of grids used in each solver - both in the horizontal and vertical directions. A sketch might be helpful too.

Regarding the reference to Grid 1 and Grid 5 starting at line 291, in the preceding paragraph at line 282-284, we state that ‘MRAMS was run using an interactive nested grid system, which is used to increasingly focus the resolution of the model from a hemispheric scale down to the region of interest using five nested domains in total.’

The grids are then referenced in regard to the size of the grids, and comparison to other similar mesoscale models. We have included an extra line of description regarding the grid resolution, stating that Grid 1 had a ‘horizontal’ grid spacing of 240 km, while grid 5 had a ‘horizontal’ grid spacing of 2 km. Details of the vertical structure are given immediately following the horizontal grid description. 

At line 284-286, we have also included ‘The MRAMS vertical levels follow the terrain topography near the surface and become increasingly horizontal as height increases above the surface’

A description of the increasing resolution toward the surface of the CFD mesh is given from line 423- 425. 

A new image of the MRAMS horizontal grid has been added (Figure 2 below)

3. The quality of the Figures is not very good, at least in the version that I see as a reviewer. They are blurry and it is almost impossible to read the text inside the figures.

The resolution of Figure 11, 12 and 13 has been increased. As far as we can see, there is no issue with other figure’s resolution.

4. In addition to the low quality of embedded figures, the discussions are either too brief or confusing. For example, Figure 1 is supposed to illustrate the workflow of the study, but no discussion of this figure is included, except for a single sentence that cites it. The same comment applies to most of the figures as well.

In regard to Figure 1, we have reworded the introduction (Line 187-192)

‘We aim to provide a new set of protocols to perform realistic microscale CFD modelling at a sub-dune length scale. This requires using the output of a GCM to inform mesoscale modelling over the area of interest, the output of which will be used to inform the initial state and boundary conditions of the CFD model. This process is outlined in Fig 2, which provides a general overview of the methodological process used for this study’

After the Figure has been inserted, extra description has been added from Line 196-202

‘Alongside the modelling workflow shown in Fig 1, the process of site selection is shown. While the region of interest can be selected from low resolution imagery, the selected site must also have temporal HiRISE imagery. The high resolution imagery from the HiRISE camera is vital for the creation of a Digital Terrain Model (DTM) which is used to create the stereolithography (STL) file. The STL file provides realistic dune topography within the CFD domain, which is crucial to properly examine wind flow behaviour over the site of interest.’

Each of the other figures have been addressed, and where appropriate, further discussion of the figures have been included, listed below

Fig 2 (New Image) is mentioned once regarding the MRAMS grid setup

Fig 3 is mentioned multiple times and is explained in detail in the figure description.

Fig 4 is a visual addition of a barchan dune shape and does not require further mention.

Fig 5 shows the CFD domain, which is described in detail, and does not need expanded on.

Fig 5 is a description of the winds and is described in text.

Fig 7 (a, and b) are mentioned 8 times in text, with supporting image (Fig 8)

Fig 8 is a supporting image for Fig 7. 

Fig 9 has now been mentioned five times in the manuscript. Previously a brief mention was made that our results (Fig 11) showed agreement with Fig 9. 

At line 645-649 we have now included: ‘Fig 12 highlights the previous statement that the wind flow profile from Durán et al. (2007) (Fig 9) is well reproduced. Both images show flow compression on the stoss slope, followed by a significant region of recirculation in the lee of the dune, and Fig 12 also shows the highlighted separation bubble between the horns of the dune seen in Fig 9.’

Fig 10 was mentioned nine times in the manuscript, both in the context of reproducing flow characteristics as well as being a reference for other images. Another three references have been added to the text regarding this image: 

Line 530-531 ‘The wind flow dynamics of Dune 1 (Fig 10) are shown in Fig 11. The high wind speeds calculated in Fig 11…’

Line 532 - 533 ‘As seen in Fig 12, for the smaller barchan dune (Dune 2 Fig 10), the maximum speed…’

Line 557 ‘The flow dynamics of Dune 2 (Fig 10) are further shown in Fig 13’

Fig 11 is mentioned seven times in the text, with particular reference to the wind flow steering

Fig 12 is mentioned eight times in the text, with reference to both wind speeds and flow steering

Fig 13 was mentioned three times in text, however further attention has been drawn to the flow distortion observed in this region:

At line 649-652 we have now included ‘The complex wind flow pattern in the lee of the dunes is similarly seen in Fig 13 (Dune 2 Fig 10). The reverse view of the dune shows both the recirculation between the horns of the dunes, but also the significant flow distortion in this region, which does not rejoin the main body of flow further downwind.’

Although this dune is not extensively mentioned, the detail of the image is mentioned in regard to previous author’s findings regarding both wind speed and direction, and now discusses the effect of flow distortion in the lee of the dunes.

With the new additions, particularly regarding Figure 1 and Figure 13, we feel that each image has been appropriately included in text and explained why some images are mentioned just once.

5. In Figure 9: the authors mention that “The arrow across the central barchan shows the path taken by the mesh independence check”. However, it is not clear what this actually means.

Now Figure 10

At line 527-529 in the figure description, we have added ‘The arrow across the central barchan shows the orientation and length of the domain probed for the mesh independence check’. 

In the explanation of the ‘Mesh independence study’ at line 475-477 states that ‘Wind speeds 1.9 m above the surface of the STL were taken over 20 points along the profile of a large barchan dune’. Therefore, we outline the path taken by the mesh independence check.

6. In Figure 9: the colorbar is only labeled at the 0 and 29.5 marks. It is really hard to interpret the values of the wind speed over the whole domain.

The maximum wind speed of this map was incorrect. This has been corrected to the proper values, and now includes further wind speed classifications

7. The authors apply the CFD model to resolve the wind flow at regions where no data is available. Thus, the only way to validate the results was by making only “qualitative” assessments. However, it is customary to first validate the solver by applying it to regions where data is available so that more quantitative and detailed assessments could be done. After that, the solver can be trusted in regions with less or no data.

This study aimed to capture microscale wind flow patterns over a barchan dune field on Mars. A combined microscale modelling approach was required as the lower resolution of GCMs and mesoscale modelling cannot fully capture the role of localised topography. Unfortunately, this site, and others like it which are controlled by secondary flow patterns over dunes, do not have in situ data. Furthermore, at sites where in situ data is available, it has been affected by a wide range of issues (see lines 73-89). Where data is available from Jezero crater and Gale crater, it is significantly impacted by the intra crater topography. This topography produces a highly complex upwind boundary layer that is not sufficiently modelled by mesoscale and macroscale modelling. This makes the sites unsuitable for the approach taken in this study

In summary, the reviewer concludes covers an interesting topic with a valid modeling approach. However, the manuscript lacks proper description of the computational setup, numerical schemes, and multi-scale coupling algorithms which makes it hard to comprehend the proposed modeling framework. In addition, the low quality of the figures and insufficient/ambiguous discussions hinder drawing insights about the physics of near-surface martian wind flow at the sub-dune length scales.

General Comments:

1) There is no reason why you would not be able to model the terra incognita region with OpenFoam, so I am not sure why this region was excluded. Would you need more computational power so that the mesh size is sufficiently small to reproduce the complex turbulence on the lee side of the Barchan?

In this study, it is at the mesoscale level where the terra incognita issue is initially encountered. This could theoretically affect the inputs to the CFD modelling, which is why specific attention is drawn to the details of this region, and other studies which have encountered this issue are referenced. 

As originally mentioned, from line 394 to 396, reference was given to the use of a 10 km boundary which did not significantly change the results of the 450 m domain. 

2) Are you coupling the models offline or using the ESMF? I would have thought you could use ESMF for this problem and it possibly would make the workflow more straightforward.

The models are coupled offline. As this is a Mars based study with vastly different atmospheric properties, we are not using the ESMF to do this, though attention has been brought to the extensive work done on Earth and why that cannot be directly implemented on Mars. Future studies may incorporate a coupled model, specifically for the GCM-to-MRAMS transition however this is beyond the scope of this paper. Even if the GCM to MRAMS setup was coupled, the MRAMS-to-CFD mapping is highly customized as outlined in the paper, which would not be appropriate for a coupled framework such as ESMF.

line 362: I would say "Turbulent kinetic energy" not "Turbulence kinetic energy"

Now Line 411, the reviewer’s comment has been amended in text.

line 370-371: "is set" not "is equivalent"

Now Line 421, the reviewer’s comment has been amended in text.

line 371: replace "k(z) and epsilon(z) are the wind speeds at height z" with "z is the height of interest"

Now Line 421, the reviewer’s comment has been amended in text.

line 399: replace "simulation was simulated" with "simulation was run"

Now Line 428, the reviewer’s comment has been amended in text.

line 381: "and in the time following" is not clear, please revise.

Now Line 429-431, this has been amended to ‘The simulation was run for a further 50 seconds to examine the CFD output. The additional 50 seconds was included to ensure there was no significant difference in wind speed results between the time taken for the wind flow to traverse the domain (the first 185 seconds) and in the following 50 seconds.’

line 393: replace "more, properly" with "more. Properly"

Now Line 444, the reviewer’s comment has been amended in text.

line 395: replace "beyond what our present investigation required" with "beyond the scope of the present investigation"

Now Line 446-447, the reviewer’s comment has been amended in text.

line 400: "immediate locale" not clear, please revise.

Now Line 451-454, this has been amended to ‘we saw little observable difference between adjacent grid points upwind of the selected region of interest shown in Fig 2c (unlike elsewhere in the caldera), and our present study does not require multiple or complex lateral inlets to the CFD domain.’

line 408. "73 of the 254 total time steps" are we talking about time steps, of about "wind states" as line 410 suggests? if you are seeing diurnal patterns, I guess your simulations are run for at least a couple of days but I am not sure I saw the period over which you are running the codes in the paper.

The mesoscale model setup is described from page 12 to page 14, where it is stated that the simulation is run for 4.5 sols, of which 3.5 have useable data due to the spin up issue associated with large scale numerical modelling. At this point, we are simply outlining that although there is diurnal variation (as is expected) and this is accounted for by averaging only those winds which were associated with sediment transport. Averaging is an appropriate method given the wind rose at this season, and to perform 73 individual CFD simulations, then average that output is not a feasible approach.

At line 462, we have reworded ‘these wind states’ to ‘the 73 individual time steps’ for further clarification.

line 552: "on Mars" not "on the Mars"

Now Line 607, the reviewer’s comment has been amended in text.

line 604: remove "more-informed"

Now Line 669, the reviewer’s comment has been amended in text.

---

## [Editor Report · Decision Letter 1]

10 Oct 2022

Full Title: From Macro- to Microscale: A combined modelling approach for near-surface wind flow on Mars at sub-dune length-scales

PONE-D-22-22872R1

Dear Dr. Love,

We’re pleased to inform you that your manuscript has been judged scientifically suitable for publication and will be formally accepted for publication once it meets all outstanding technical requirements.

Kind regards,

Omer San

Academic Editor

PLOS ONE

Additional Editor Comments (optional):

All issues and concerns raised in reviewers' earlier feedback have been adequately addressed by the authors. I am mostly satisfied with the responses/revision. Therefore, I support publication of this manuscript.

---

## [Editor Report · Acceptance letter]

13 Oct 2022

PONE-D-22-22872R1 

From  Macro-  to  Microscale:  A  combined  modelling  approach  for  near-surface  wind  flow  on  Mars  at  sub-dune  length-scales 

Dear Dr. Love:

I'm pleased to inform you that your manuscript has been deemed suitable for publication in PLOS ONE. Congratulations! Your manuscript is now with our production department. 

Kind regards, 

on behalf of

Dr. Omer San 

Academic Editor

PLOS ONE